# Urinary C5b-9 as a Prognostic Marker in IgA Nephropathy

**DOI:** 10.3390/jcm11030820

**Published:** 2022-02-03

**Authors:** Byung Chul Yu, Jin Hoon Park, Kyung Ho Lee, Young Seung Oh, Soo Jeong Choi, Jin Kuk Kim, Moo Yong Park

**Affiliations:** Division of Nephrology, Department of Internal Medicine, Soonchunhyang University Bucheon Hospital, 170 Jomaru-ro, Bucheon 14584, Korea; nephroybc@schmc.ac.kr (B.C.Y.); wlsgns0620@naver.com (J.H.P.); futurelkh@schmc.ac.kr (K.H.L.); 108254@schmc.ac.kr (Y.S.O.); crystal@schmc.ac.kr (S.J.C.); medkjk@schmc.ac.kr (J.K.K.)

**Keywords:** complement system proteins, IgA nephropathy, membrane attack complex, prognostic marker, urinary C5b-9

## Abstract

C5b-9 plays an important role in the pathogenesis of immunoglobin A nephropathy (IgAN). We evaluated C5b-9 as a prognostic marker for IgAN. We prospectively enrolled 33 patients with biopsy-proven IgAN. We analyzed the correlation between baseline urinary C5b-9 levels, posttreatment changes in their levels, and clinical outcomes, including changes in proteinuria, estimated glomerular filtration rate (eGFR), and treatment response. Baseline urinary C5b-9 levels were positively correlated with proteinuria (r = 0.548, *p* = 0.001) at the time of diagnosis. Changes in urinary C5b-9 levels were positively correlated with changes in proteinuria (r = 0.644, *p* < 0.001) and inversely correlated with changes in eGFR (r = −0.410, *p* = 0.018) at 6 months after treatment. Changes in urinary C5b-9 levels were positively correlated with time-averaged proteinuria during the follow-up period (r = 0.461, *p* = 0.007) but were not correlated with the mean annual rate of eGFR decline (r = −0.282, *p* = 0.112). Baseline urinary C5b-9 levels were not a significant independent factor that could predict the treatment response in logistic regression analyses (odds ratio 0.997; 95% confidence interval, 0.993 to 1.000; *p* = 0.078). Currently, urinary C5b-9 is not a promising prognostic biomarker for IgAN, and further studies are needed.

## 1. Introduction

Immunoglobin A nephropathy (IgAN) is the most common cause of primary glomerulonephritis (GN) worldwide and is characterized by the deposition of IgA in the glomerular mesangium [1]. The pathogenesis of IgAN has been described as a multi-hit process [2]. In patients with IgAN, underglycosylated IgA1 (Gd-IgA1), the production of which is thought to be caused by environmental and genetic factors, is bound by autoantibodies (auto-Abs) [3]. Gd-IgA1-auto-Ab immune complexes activate the complement system and lead to glomerular inflammation, mesangial proliferation, and ultimately, fibrosis in patients with IgAN [4]. Disease clinical course ranges from asymptomatic hematuria to chronic kidney disease. While patients preserve their kidney function during their lifetime, 6% to 43% progress to end-stage kidney disease within 10 years after diagnosis [5,6]. Therefore, studies are underway to develop a biomarker that can predict the prognosis of patients with IgAN, and interest in complement activation has increased recently.

Deposited Gd-IgA1-auto-Ab immune complexes activate the alternative and lectin pathways but not the classical pathway in IgAN [7]. For instance, C3, properdin, complement factor H, C5b-9 of the alternative pathway, and C4d (the mannose-binding lectin of the lectin pathway) are present in the mesangial immune deposits in kidney tissue from patients with IgAN [8,9,10]. This suggests that complement pathway activation plays an important role in IgAN pathogenesis. All three complement pathways lead to C5 convertase production, which cleaves into C5a and C5b. C5b binds to C6, C7, C8, and C9, producing C5b-9 [7]. C5b-9, also known as the membrane attack complex, is the end-product of the three complement pathways via this terminal complement cascade [11]. C5b-9 forms a pore through the cell membrane and disrupts cell integrity, and causes subsequent cell lysis. The C5b-9 formation induces kidney cell injury, inflammation, and fibrosis [7]. C5b-9 plays an important role in the pathogenesis of various GN, including membranous nephritis [12], membranoproliferative GN [13], lupus nephritis [14], and C3 glomerulopathy [15]. Moreover, previous studies have shown that C5b-9 is associated with pathogenesis and kidney injury in patients with IgAN [16,17].

We hypothesized that C5b-9 might serve as a valuable biomarker for predicting treatment outcomes in IgAN. We measured baseline C5b-9 levels at the time of diagnosis and changes in C5b-9 levels after medical treatment and assessed their correlation with clinical outcomes, including treatment response, changes in proteinuria, and estimated glomerular filtration rate (eGFR) in patients with IgAN.

## 2. Materials and Methods

### 2.1. Study Population

We prospectively screened biopsy-proven patients with IgAN from the Cohort for Biomarker Inquiry of Renal Aggravation (COBRA), which collects clinical information, kidney tissue, serum, and 24-hour-collected urine samples from patients with GN at Soonchunhyang University Seoul, Bucheon, and Cheonan Hospitals from January 2016 to December 2019. A pathologist who was not involved in the initial diagnosis reviewed the pathological findings of all patients, including the Oxford classification. Of the total 125 patients with IgAN enrolled in this cohort, patients with less than 8 glomeruli used for diagnosis (*n* = 27), uncertain IgAN diagnosis (*n* = 12), or other concomitant diseases (such as hypertensive nephrosclerosis, diabetic nephropathy, tubulointerstitial nephritis, and focal segmental glomerulosclerosis (*n* = 25)) were excluded from the study. We excluded patients with an eGFR < 30 mL/min/1.73 m^2^ (*n* = 8) using the Chronic Kidney Disease Epidemiology Collaboration (CKD-EPI) equation. Among the remaining patients, 33 patients, from whom serum and urine samples at the time of diagnosis and 6 months after diagnosis were available and who had accurate clinical data recorded periodically at least every 3 months for more than 1 year after the diagnosis, were finally enrolled and analyzed.

We classified patients into response and non-response groups based on the treatment response. The response group was defined by proteinuria < 0.3 g per 24 h or a decrease in proteinuria by at least 50% from the baseline value at the time of kidney biopsy and <3.5 g per 24 h at 6 months after medical treatment according to Kidney Disease Improving Global Outcomes (KDIGO) guidelines [18].

### 2.2. Clinical and Laboratory Data Collection

We collected data on patient’s demographics and comorbidities, including diabetes and hypertension history, and measured body mass index and mean arterial pressure (MAP) at the time of kidney biopsy. Laboratory data were obtained at every visit during the planned study period. The eGFR levels were determined from serum creatinine values using the CKD-EPI equation. Proteinuria levels were obtained by 24-hour urine collection, and time-averaged proteinuria (TA proteinuria) was calculated as the average of the mean of proteinuria measurements every 6 months for each patient after exclusion of baseline values. We reviewed the pathological findings from kidney tissues at the diagnosis and specifically checked the pathological severity according to the Oxford classification [19]. The type, duration, and total dose of administered medications, including immunosuppressive agents and angiotensin II receptor blockers (ARBs) during the follow-up period were reviewed.

### 2.3. Serum and Urinary C5b-9 Quantification

Serum and 24-hour urine samples were obtained from subjects at the time of diagnosis and 6 months after medical treatment, stored at −80°C until use, and were thawed to 37 °C immediately before the assay. Serum and urinary C5b-9 concentrations were measured using commercially available sandwich enzyme-linked immunosorbent assay kits (SC5b-9 EIA; Quidel, San Diego, CA, USA), according to the manufacturer’s instructions. C5b-9 values were expressed as absorbance values (OD 450 nm) after subtraction of background absorbance from human serum control samples [17,20,21]. To minimize the effects of urine volume, urinary C5b-9 levels were relativized to the urine creatinine concentration of the same sample and expressed as corrected urinary C5b-9 levels. A single investigator, blinded to the clinical information, performed all measurements.

### 2.4. Statistical Analysis

Descriptive characteristics of the study population are reported as mean ± standard deviation for continuous variables and as frequency counts with percentages for categorical and binary variables. Comparisons between groups were made using Mann–Whitney and Wilcoxon signed-rank tests for continuous variables and either chi-squared test or Fisher’s exact tests for categorical variables, as appropriate. Serum and corrected urinary C5b-9 levels between groups were compared using the Mann–Whitney test. Spearman’s rank correlation coefficient was used to analyze the relationships between serum and corrected urinary C5b-9 levels and clinical and histological variables. All statistical tests were two-sided, and *p*-values lower than 0.05 were considered statistically significant. All analyses were performed using SPSS 25 for Windows (SPSS Inc., Chicago, IL, USA) or Graphpad Prism5 (GraphPad Software, La Jolla, CA, USA).

## 3. Results

### 3.1. Study Population

Patient’s baseline characteristics are presented in Table 1. All patients were treated with ARBs as a renin–angiotensin–aldosterone system (RAAS) blockade. All patients included in this study were treated with ARBs from baseline to six months. They continued to receive ARBs during the follow-up period, although some patients temporarily discontinued ARBs due to well-known adverse effects of the latter, such as a decrease in blood pressure and eGFR, and the occurrence of hyperkalemia. As a result of reviewing the medical records of all patients enrolled in this study, they received corticosteroids as immunosuppressive treatment if they showed proteinuria of 1 g/day or more after 3 to 6 months of ARBs treatment. Combination therapy with ARBs and corticosteroids as immunosuppressive agents was administered to 21.2% of the enrolled patients. At the time of diagnosis, the mean number of glomeruli obtained from enrolled patients’ kidneys was 23.1 ± 15.6, with 12.1% and 8.8% corresponding to global and segmental sclerosis, respectively. Enrolled patients were classified into the response group (*n* = 23) and the non-response group (*n* = 10). Baseline characteristics such as age, sex, MAP, eGFR, proteinuria amount, and Oxford classification did not differ between the two groups. Furthermore, there were no significant differences in baseline serum and corrected urinary C5b-9 levels between the response and non-response groups (Table 1).

### 3.2. Baseline-Corrected Urinary C5b-9 Levels Were Correlated with MAP, Proteinuria Amount, and Segmental Sclerosis at the Time of Diagnosis

There was no correlation between baseline serum and corrected urinary C5b-9 levels (r = −0.115, *p* = 0.612). Baseline serum C5b-9 levels did not correlate with conventional prognostic markers in IgAN as MAP, eGFR, proteinuria amount, or Oxford classification at the time of diagnosis [1] (Table 2). Baseline-corrected urinary C5b-9 levels correlated with MAP and the proteinuria amount (Table 2). Baseline serum and corrected urinary C5b-9 levels were lower in patients with segmental sclerosis according to the Oxford classification (Appendix A), and were negatively correlated with the proportion of glomeruli with segmental sclerosis and the sum of glomeruli with global and segmental sclerosis among glomeruli included in kidney tissue samples at the time of diagnosis (Appendix A).

### 3.3. Changes in Corrected Urinary C5b-9 Levels Were Positively Correlated with Changes in Proteinuria and Inversely Correlated with Changes in eGFR at 6 Months after Medical Treatment

Changes in corrected urinary C5b-9 levels were positively correlated with changes in proteinuria and inversely correlated with changes in eGFR at 6 months after medical treatment. Changes in serum C5b-9 levels did not correlate with changes in proteinuria or eGFR (Figure 1). Changes in serum and corrected urinary C5b-9 levels were not correlated with the MEST-C score of the Oxford classification at the time of diagnosis (Appendix A).

### 3.4. Corrected Urinary C5b-9 Levels Were Decreased in the Response Group 6 Months after Medical Treatment

While corrected urinary C5b-9 levels decreased in the response group but did not in the non-response group at 6 months after medical treatment (Figure 2), serum C5b-9 levels did not decrease in any group. In the response group, corrected urinary C5b-9 levels decreased in both patients receiving ARBs alone and corticosteroids in combination with ARBs (Figure 3).

Based on logistic regression analyses between the prognostic value of baseline-corrected C5b-9 levels with those of traditional prognostic markers, including eGFR, proteinuria, and hypertension, no statistically significant independent factors could be identified that could predict the treatment response (Table 3).

### 3.5. Changes in Corrected Urinary C5b-9 Levels Were Positively Correlated with Time-Averaged Proteinuria but Were Not Correlated with the Mean Annual Rate of eGFR Decline during Follow-Up Duration

The mean and median follow-up duration of enrolled patients were 3.4 ± 0.8 and 4.0 years, respectively. Mean TA proteinuria and annual rate of eGFR decline were 518.7 ± 527.3 mg/day and 0.07 ± 6.99 mL/min/1.73 m^2^, respectively. Changes in corrected urinary C5b-9 levels were positively correlated with TA proteinuria but were not correlated with the mean annual rate of eGFR decline during follow-up duration. Baseline serum and corrected urinary C5b-9 levels and changes in serum C5b-9 levels did not correlate with either TA proteinuria or the mean annual rate of eGFR decline (Table 4).

## 4. Discussion

The current study evaluated C5b-9 efficacy as a prognostic marker for IgAN. Several studies on C5b-9 deposition in IgAN showed that all or almost all patients with IgAN had more intense and diffuse C5b-9 deposition in the glomerulus and tubules compared with healthy controls, and its deposition sites were colocalized with IgA and C3-containing immune complexes [7,22,23]. Besides, C5b-9 staining intensities were positively correlated with serum creatinine levels [16,24] and the proteinuria amount [25,26]. Kidney biopsy is an invasive procedure and, thus, repeated testing is difficult. Because urine samples are not invasive and can be obtained repeatedly during the follow-up period, urinary biomarkers have the advantage of being easy to analyze. Various studies are being conducted to discover and validate urinary biomarkers that can predict IgAN prognosis [27,28,29,30]. In this context, recent studies on the efficacy of urinary C5b-9 as a biomarker in IgAN were conducted, and urinary C5b-9 levels increased in patients with IgAN compared with healthy volunteers and were associated with disease severity and inversely correlated with eGFR in patients with IgAN [16,24]. Since these studies were cross-sectional, they could not evaluate changes in urinary C5b-9 levels and their relevance to clinical outcomes after treatment. The current longitudinal study was conducted to not only evaluate the relationship between baseline urinary C5b-9 levels and disease severity at the time of diagnosis but also changes in urinary C5b-9 levels and clinical outcomes after treatment in patients with IgAN.

We observed that baseline urinary C5b-9 levels and changes in urinary C5b-9 levels were positively correlated with the proteinuria amount at the time of diagnosis and with TA proteinuria during follow-up, respectively. Hence, our findings show a close relationship between urinary C5b-9 levels and proteinuria. Because the current study was an observational study, a causal relationship between proteinuria and urinary C5b-9 levels remained unknown. Therefore, the relationship between complement and proteinuria can only be inferred from previous research results. Previous experimental studies have shown that an imbalance between complement and complement regulatory proteins induces podocyte membrane insertion of C5b-9, which leads to various intracellular responses [31,32]. This process results in the glomerular basement membrane degradation and proteinuria and glomerular sclerosis development [12]. C5b-9 inserted into the podocyte membrane is transported intracellularly and extruded into the urinary space; it appears in the urine [33]. Thus far, no research suggests that an increase in proteinuria increases the complement protein to date; hence, it is more reasonable to consider urinary C5b-9 levels rather than proteinuria as a causative factor in the positive correlation between urinary C5b-9 levels and proteinuria.

Baseline urinary C5b-9 levels were inversely correlated with the percentage of segmental sclerosis as a fraction of all glomeruli and lower in patients with segmental sclerosis according to the Oxford classification. The presence of segmental sclerosis was observed in 57.6% of enrolled patients, like other study cohorts reporting 66% to 76% [34,35,36]. Although the exact segmental sclerosis pathogenesis is still unclear, it is thought to be caused by post-inflammatory scarring of segmental proliferative or necrotizing lesions, adaptive responses following nephron loss, and podocyte damage induced by mediators released from mesangial cells [36,37]. These theories all show that segmental sclerosis is a subacute or chronic change caused by inflammation rather than the state in which an acute inflammatory response actively occurs. Patients with a higher proportion of segmental sclerosis may reflect that they are in a subacute or chronic phase following an active inflammatory phase involving complement pathway activation. Therefore, the lower urinary C5b-9 levels in these patients seem to be consistent with these hypotheses. However, to clarify this assumption, an experimental study on the change in the complement pathway activity during the occurrence and progression of segmental sclerosis during IgAN course is needed.

We analyzed the response to treatment, changes in proteinuria and eGFR six months after medical treatment, and TA proteinuria and mean annual rate of eGFR decline during follow-up duration as clinical outcomes. These outcomes were established based on the results of several previous discussions and studies on surrogate outcomes in clinical trials that can serve as substitutes for definitive clinical outcomes. The 2018 US Food and Drug Administration workshops on clinical trials in CKD support changes in GFR, difference in GFR slopes, and a reduction in mean albuminuria as valid surrogate outcomes for CKD progression [38]. In addition, a recent study showed that the combination of a decrease in eGFR and an increase in urinary albumin–creatinine ratio predicted the outcomes of advanced CKD better than either alone [39]. In the current study, we observed that urinary C5b-9 levels were associated with proteinuria. A negative correlation was observed between changes in urinary C5b-9 levels and changes in eGFR six months after medical treatment. Considering that a recent study provides a rigorous foundation for the use of proteinuria reduction as an endpoint in IgAN [40], although urinary C3b-9 levels were associated with a decrease in proteinuria alone, urinary C5b-9 levels might be considered as a prognostic factor in IgAN. Although there was no statistical significance, baseline and changes in urinary C5b-9 levels tended to be negatively correlation with baseline eGFR and the mean annual rate of eGFR decline, respectively. A decrease in proteinuria typically occurred within months after treatment, whereas demonstration of differences in eGFR decline required much longer follow-up in studies on the clinical outcomes of IgAN [40,41]. The mean follow-up duration of enrolled patients in the current study was 3.4 years; however, it may have been too short to establish an association between urinary C5b-9 levels and eGFR. The correlation coefficient between baseline-corrected urinary C5b-9 levels and eGFR was −0.236. The result of the power calculation using this method was 24.7%. Considering this result, it is presumed that the number of analyzed subjects was too small to show a significant correlation. Although it showed higher significance than the other conventional risk factors, it was not a significant independent factor affecting treatment response. Considering that conventional risk factors were also not significant, it is thought that this was also because the sample size was too small for logistic regression analyses. If further research is conducted with larger cohorts and long-term follow-up durations, a significant relationship between urinary C5b-9 levels, eGFR, and treatment response may be obtained, and the current study might be used as a pilot study for such future studies.

When compared to the response and non-response groups regarding to changes in proteinuria after initial treatment, urinary C5b-9 levels decreased in the response group, but there was no significant change in the non-response group. In the response group, urinary C5b-9 levels decreased in both patients receiving ARBs alone and corticosteroids in combination with ARBs. These results may be interpreted in two ways. First, the current treatment strategies, including RAAS blockade with and without glucocorticoid, may directly or indirectly inhibit the complement pathway activation, and this inhibitory effect may differ from patient to patient. However, considering that changes in urinary C5b-9 levels only correlate with changes in eGFR at 6 months after treatment, current treatment strategies seem to be suboptimal in attenuating the long-term effect of complement pathway activation. Second, regardless of treatment, there are patients whose complement pathway may be activated during the active phase of the disease and then spontaneously attenuated, while patients whose complement pathway does not attenuate and continue to be active. Since the current study was an observational study, the causal relationship cannot be established and additional studies are needed to confirm changes in complement pathway activation during the IgAN disease course to verify these hypotheses.

There were several limitations to the current study. One of the major limitations is the small number of subjects enrolled and analyzed in the current study. The authors tried to analyze only pure IgAN patients with more reliable clinical information compared to previous studies. To minimize the effect of other concomitant diseases, including diabetic and hypertensive nephropathy, and exclude an indeterminately diagnosed IgAN, approximately only 70% of patients with IgAN from the COBRA cohort were included. Among these patients, as patients with unclear clinical data or follow-up loss were excluded from the analysis, the number of enrolled and analyzed patients was smaller than in other previous studies. Contrary to the result of previous studies [16,17], there was a negative correlation between baseline urinary C5b-9 levels and eGFR, but this did not reach statistical significance, possibly because of the small sample size. Second, the mean follow-up duration of 3.4 years was too short to evaluate the long-term clinical outcomes of patients with IgAN. This may explain why there was no significant association between changes in urinary C5b-9 levels and the mean annual rate of eGFR decline. Third, another limitation of our study is that we did not compare C5b-9 immunohistochemistry staining intensity in kidney tissue from enrolled patients at the time of diagnosis with urinary C5b-9 levels. The current study was conducted on the premise that there would be a positive correlation between the renal C5b-9 activity degree and urinary C5b-9 levels. However, urinary C5b-9 levels may increase as systemic C5b-9 levels increased by other clinical factors, and filtration into the urine cannot be excluded. A significant decline in GFR may decrease urinary C5b-9 levels. To minimize these limitations, we confirmed no correlation between serum and urine C5b-9 levels and excluded patients with advanced CKD with an eGFR lower than 30 mL/min/1.73 m^2^. Finally, the C5b-9 levels after treatment were measured only once at 6 months after treatment. Although we were able to analyze whether changes in urinary C5b-9 levels after short-term treatment could be used as a biomarker for predicting long-term outcomes after treatment, periodic measurements of urinary C5b-9 levels are required to confirm the exact pattern of changes in urinary C5b-9 levels during treatment. To overcome this limitation, additional studies evaluating the correlation between urinary C5b-9 levels at 1 and 3 years after treatment and clinical outcomes are underway.

## 5. Conclusions

In conclusion, the current study showed that urinary C5b-9 levels are associated with short- and long-term clinical outcomes in patients with IgAN. However, currently urinary C5b-9 is not a promising prognostic biomarker for IgAN. Further studies with larger cohorts and long-term follow-up durations are needed to clarify this.

## Figures and Tables

**Figure 1 jcm-11-00820-f001:**
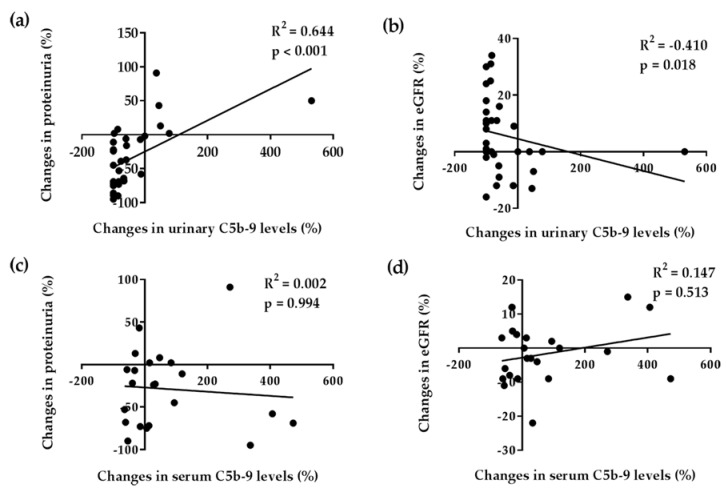
Relationships between changes in serum and urinary complement C5b-9 levels, proteinuria, and estimated glomerular filtration rate (eGFR) after medical treatment. Changes in urinary levels of C5b-9 positive correlate with changes in proteinuria (**a**) and inversely correlate with changes in eGFR (**b**) at 6 months after medical treatment. Changes in serum levels of complement C5b-9 does not correlate with changes in proteinuria (**c**) and eGFR (**d**). Data were analyzed by Spearman’s rank correlation coefficient.

**Figure 2 jcm-11-00820-f002:**
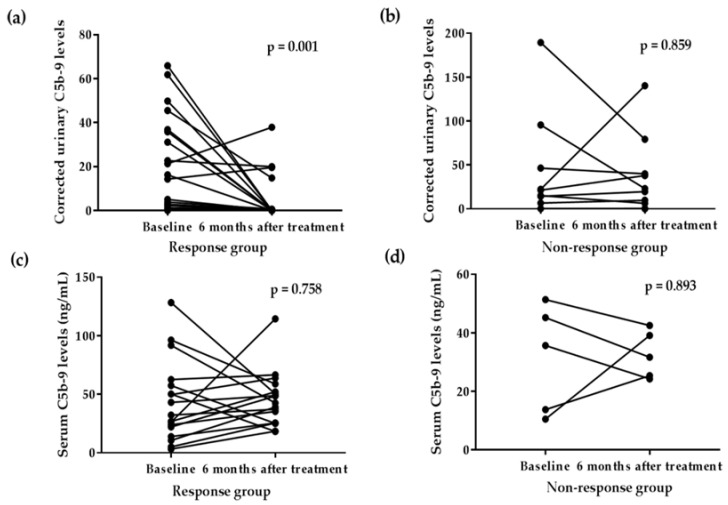
Changes in serum and urinary levels complement C5b-9 levels in response and non-response group at 6 months after medical treatment. While corrected urinary C5b-9 levels decreased in the response group (**a**) but did not in the non-response group (**b**). Serum C5b-9 levels did not decrease in both response (**c**) and non-response group (**d**). The response group was defined as proteinuria <0.3 g per 24 h or a decrease in proteinuria by at least 50% from the initial value and <3.5 g per 24 h at 6 months after medical treatment. Data were analyzed by Wilcoxon matched-pairs signed-rank tests.

**Figure 3 jcm-11-00820-f003:**
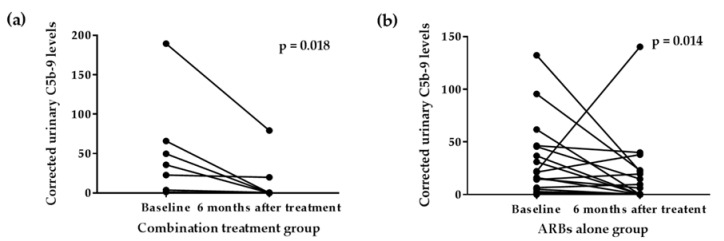
Changes in urinary complement C5b-9 levels in (**a**) combination treatment group (patients who were treated with angiotensin receptor blockers (ARBs) and immunosuppressive agent) and (**b**) ARBs alone group at 6 months after medical treatment. Data were analyzed by Wilcoxon matched-pairs signed-rank tests.

**Table 1 jcm-11-00820-t001:** Baseline characteristics of enrolled patients.

Variable	Total (*n* = 33)	Response Group (*n* = 23)	Non-Response Group (*n* = 10)	*p*-Value *
Sex (male)	17 (51.5)	10 (43.5)	7 (70.0)	0.259
Age (years)	40.0 ± 14.4	41.0 ± 15.2	37.5 ± 12.8	0.603
Body mass index (kg/m^2^)	24.3 ± 4.1	23.9 ± 4.0	25.2 ± 4.6	0.524
Mean arterial pressure (mmHg)	93.4 ± 11.9	92.4 ± 12.2	95.8 ± 11.4	0.384
Hypertension	5 (15.2)	4 (17.4)	1 (10.0)	>0.999
Serum creatinine levels (mg/dL)	1.12 ± 0.29	1.09 ± 0.30	1.17 ± 0.26	0.324
eGFR (mL/min/1.73 m^2^)	79.76 ± 22.67	79.38 ± 23.18	80.65 ± 22.62	0.773
Proteinuria (g/24 h)	1.05 ± 0.97	1.14 ±1.09	0.85 ± 0.60	>0.999
Serum C5b-9 levels (ng/mL)	43.19 ± 31.55	43.69 ± 34.13	31.31 ± 18.40	0.493
Corrected urinary C5b-9 levels	283.22 ± 421.22	227.61 ± 321.55	411.12 ± 593.33	0.499
Immunosuppressive agents	7 (21.2)	6 (26.1)	1 (10.0)	0.397
Oxford classification				
M score 1 ^1^	17 (51.5)	10 (43.5)	7 (70.0)	0.259
E score 1 ^2^	10 (30.3)	8 (34.8)	2 (20.0)	0.682
S score 1 ^3^	19 (57.6)	13 (56.5)	6 (60.0)	>0.999
T score 1–2 ^4^	11 (33.3)	8 (34.8)	3 (30.0)	>0.999
C score 1–2 ^5^	4 (12.1)	4 (17.4)	0 (0.0)	0.289
Global sclerosis (%)	12.1 ± 11.9	12.7 ± 11.4	10.6 ± 13.5	0.499
Segmental sclerosis (%)	8.8 ± 11.9	9.1 ± 12.8	8.0 ± 10.4	0.985

Data are presented as mean ± standard deviation for continuous variables or number (%) for categorical variables. eGFR, estimated glomerular filtration rate. ^1^ Mesangial hypercellularity score > 0.5; ^2^ endocapillary proliferation = present; ^3^ segmental glomerulosclerosis/adhesion = present; ^4^ severity of tubular atrophy/interstitial fibrosis (T1 = 26–50%; T2 > 50%); ^5^ presence of crescent (C1 = 1–25%; C2 = 26–100%). * The *p*-value was used to indicate the differences between the response and non-response groups.

**Table 2 jcm-11-00820-t002:** Relationship between baseline serum and corrected urinary C5b-9 levels and conventional risk factors at presentation in patients with IgA nephropathy.

Variable	Mean Arterial Pressure	eGFR	Amount of Proteinuria
Serum C5b-9 levels (ng/mL)	r = −0.020	r = −0.156	r = −0.078
	*p* = 0.930	*p* = 0.487	*p* = 0.728
Corrected urinary C5b-9 levels	r = 0.489	r = −0.236	r = 0.548
	*p* = 0.004	*p* = 0.185	*p* = 0.001

Data were analyzed by Spearman’s rank correlation coefficient. eGFR, estimated glomerular filtration rate.

**Table 3 jcm-11-00820-t003:** Univariable and multivariable logistic regression analyses of independent prognostic factors for treatment response.

Risk Factor	Univariable		Multivariable	
	OR (95% CI)	*p*-Value	OR (95% CI)	*p*-Value
Baseline-corrected urinary C5b-9 levels	0.999 (0.997–1.001)	0.270	0.997 (0.993–1.000)	0.078
eGFR at baseline (mL/min/1.73m^2^)	0.998 (0.965–1.031)	0.880	1.011 (0.968–1.055)	0.631
Proteinuria at baseline (mg/day)	1.000 (1.000–1.001)	0.435	1.001 (0.999–1.002)	0.375
Hypertension	1.897 (0.184–19.482)	0.591	67.625 (0.168–27179.9)	0.168

OR, odds ratio; 95% CI, 95% confidence interval; eGFR; estimated glomerular filtration rate.

**Table 4 jcm-11-00820-t004:** Correlation among baseline and changes in serum and corrected urinary C5b-9 levels after medical treatment.

Variable	Mean Annual Rate of eGFR Decline	Time-Averaged Proteinuria
Baseline serum C5b-9 levels (ng/mL)	r = 0.007	r = −0.215
	*p* = 0.974	*p* = 0.336
Baseline-corrected urinary C5b-9 levels	r = −0.319	r = 0.280
	*p* = 0.071	*p* = 0.115
Changes in serum C5b-9 (%)	r = 0.080	r = 0.019
	*p* = 0.725	*p* = 0.934
Changes in corrected urinary C5b-9 (%)	r = −0.282	r = 0.461
	*p* = 0.112	*p* = 0.007

Data were analyzed by Spearman’s rank correlation coefficient. eGFR, estimated glomerular filtration rate.

## Data Availability

The datasets generated and/or analyzed during the current study are available from the corresponding author upon reasonable request.

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
