# Peer review of "Urinary C5b-9 as a Prognostic Marker in IgA Nephropathy"

_jcm, 2022, doi:10.3390/jcm11030820_

Round 1

Reviewer 1 Report

Byung Chul Yu et al. studied the efficacy of urinary C5b-9 as a prognostic marker in IgA nephropathy (IgAN) patients. They found a correlation between urinary C5b-9 and proteinuria at diagnosis. Moreover, changes in urinary C5b-9 correlated with changes in proteinuria and time-averaged (TA) proteinuria. They concluded that urinary C5b-9 could be a prognostic marker in IgAN; however, the authors should address the following issues. 

Major;

1. As the authors mentioned in the manuscript, the sample size is small, and follow-up duration is short to demonstrate the role of urinary C5b-9 as prognostic marker of IgAN. Power calculation should be provided in the manuscript. 

2. Urinary C5b-9 at the time of diagnosis could not differentiate the response group from the non-response group. Also, it did not associate with mean annual rate of eGFR decline. These results provide little evidence to be a prognostic marker.

3. Changes in urinary C5b-9 were correlated with those in proteinuria and TA-proteinuria. Urinary C5b-9 could be derived partly from serum; thus, changes in urinary C5b-9 may reflect those in proteinuria irrespective of glomerular complement activation. The authors should comment on this point.

Minor;

1. Information about indications for immunosuppressive treatment of IgAN patients was not described in the manuscript. 

2. Changes in proteinuria could correlate with those in eGFR by the effect of RAS inhibitors. The authors should clarify this relationship.

Author Response

Reviewer 1

Major;

1. As the authors mentioned in the manuscript, the sample size is small, and follow-up duration is short to demonstrate the role of urinary C5b-9 as prognostic marker of IgAN. Power calculation should be provided in the manuscript. 

Thank you for your comment. In accordance with your suggestion, we have incorporated the required changes in the results section. The correlation coefficient between baseline corrected urinary C5b-9 levels and eGFR was -0.236. The result of the power calculation using this method was 24.7%. Considering this result, it is presumed that the number of analyzed subjects was too small to show a significant correlation. On the other hand, if more patients are enrolled and analyzed in a future study, a significant relationship between urinary C5b-9 levels and eGFR may be observed, and the current study might serve as a basis for such future studies. As you pointed out, further studies with larger cohorts and long-term follow-up durations are needed to clarify this.

Discussion, page 9, line 306:

“The correlation coefficient between baseline corrected urinary C5b-9 levels and eGFR was -0.236. The result of the power calculation using this method was 24.7%. Considering this result, it is presumed that the number of analyzed subjects was too small to show a significant correlation.”

2. Urinary C5b-9 at the time of diagnosis could not differentiate the response group from the non-response group. Also, it did not associate with mean annual rate of eGFR decline. These results provide little evidence to be a prognostic marker.

Thank you for your comment. I agree with your viewpoint. We incorporated a description and added related references for this issue in the discussion section.

Discussion, page 8, line 284:

“We analyzed the response to treatment, changes in proteinuria and eGFR six months after medical treatment, and TA-proteinuria and mean annual rate of eGFR decline during follow-up duration as clinical outcomes. These outcomes were established based on the results of several previous discussions and studies on surrogate outcomes in clinical trials that can serve as substitutes for definitive clinical outcomes. The 2018 US Food and Drug Administration workshops on clinical trials in CKD support changes in GFR, difference in GFR slopes, and a reduction in mean albuminuria as valid surrogate outcomes for CKD progression [38]. In addition, a recent study showed that the combination of a decrease in eGFR and an increase in urinary albumin-creatinine ratio predicted the outcomes of advanced CKD better than either alone [39]. In the current study, we observed that urinary C5b-9 levels were associated with proteinuria. A negative correlation was observed between changes in urinary C5b-9 levels and changes in eGFR six months after medical treatment. Considering that a recent study provides a rigorous foundation for the use of proteinuria reduction as an endpoint in IgAN [40], although urinary C3b-9 levels were associated with a decrease in proteinuria alone, urinary C5b-9 levels might be considered as a prognostic factor in IgAN. Although there was no statistical significance, baseline and changes in urinary C5b-9 levels tended to be negatively correlation with baseline eGFR and the mean annual rate of eGFR decline, respectively. A decrease in proteinuria typically occurred within months after treatment, whereas demonstration of differences in eGFR decline required much longer follow-up in studies on the clinical outcomes of IgAN [40, 41]. The mean follow-up duration of enrolled patients in the current study was 3.4 years; however, it may have been too short to establish an association between urinary C5b-9 levels and eGFR. The correlation coefficient between baseline corrected urinary C5b-9 levels and eGFR was -0.236. The result of the power calculation using this method was 24.7%. Considering this result, it is presumed that the number of analyzed subjects was too small to show a significant correlation. Although it showed higher significance than the other conventional risk factors, it was not a significant independent factor affecting treatment response. Considering that conventional risk factors were also not significant, it is thought that this was also because the sample size was too small for logistic regression analyses. If further research is conducted with larger cohorts and long-term follow-up durations, a significant relationship between urinary C5b-9 levels, eGFR, and treatment response may be obtained, and the current study might be used as a pilot study for such future studies.”

3. Changes in urinary C5b-9 were correlated with those in proteinuria and TA-proteinuria. Urinary C5b-9 could be derived partly from serum; thus, changes in urinary C5b-9 may reflect those in proteinuria irrespective of glomerular complement activation. The authors should comment on this point.

Thank you for your insightful comment. As you suggested, this was the most critical result of our study. We have incorporated a description of this issue in the discussion section.

Discussion, page 8, line 252:

“We observed baseline urinary C5b-9 levels and changes in urinary C5b-9 levels were positively correlated with the proteinuria amount at the time of diagnosis and with TA-proteinuria during follow-up, respectively. Hence, our findings show a close relationship between urinary C5b-9 levels and proteinuria. Because the current study was an observational study, causal relationship between proteinuria and urinary C5b-9 levels remained unknown. Therefore, the relationship between complement and proteinuria can only be inferred from previous research results. Previous experimental studies have shown that an imbalance between complement and complement regulatory proteins induces podocyte membrane insertion of C5b-9, which leads to various intracellular responses [31,32]. This process results in the glomerular basement membrane degradation and proteinuria and glomerular sclerosis development [12]. C5b-9 inserted into the podocyte membrane is transported intracellularly and extruded into the urinary space; it appears in the urine [33]. Thus far, no research suggests that an increase in proteinuria increases the complement protein to date; hence, it is more reasonable to consider urinary C5b-9 levels rather than proteinuria as a causative factor in the positive correlation between urinary C5b-9 levels and proteinuria.”

Minor;

1. Information about indications for immunosuppressive treatment of IgAN patients was not described in the manuscript. 

Per your suggestion, we have described this in the results section.

Results 3.1, page 3, line 130:

“As a result of reviewing the medical records of all patients enrolled in this study, they received corticosteroids as immunosuppressive treatment if they showed proteinuria of 1 g/day or more after 3 to 6 months of ARBs treatment.”

2. Changes in proteinuria could correlate with those in eGFR by the effect of RAS inhibitors. The authors should clarify this relationship.

Thank you for your suggestion. All patients included in this study were treated with ARBs as RAAS blockers from baseline to six months. They continued to receive ARBs during the follow-up period, although some patients temporarily discontinued ARBs due to well-known adverse effects of the latter, such as decrease in blood pressure and eGFR, and the occurrence of hyperkalemia. Therefore, analyzing the relationship between the change in eGFR according to ARBs and the change in proteinuria is characterized by certain limitations. However, as you pointed out, the effect of ARBs on proteinuria is very important; thus, we have incorporated a description of this issue in the results section.

Results 3.1, page 3, line 126:

“All patients included in this study were treated with ARBs as RAAS blockers from baseline to six months. They continued to receive ARBs during the follow-up period, although some patients temporarily discontinued ARBs due to well-known adverse effects of the latter, such as decrease in blood pressure and eGFR, and the occurrence of hyperkalemia.”

Reviewer 2 Report

In this manuscript Yu and colleagues analyzed the value of serum und urine C5b-9 levels for prediction of prognosis in 33 patients with biopsy proven IgA nephropathy. 21% of the patients were treated with corticosteroids, and the authors defined responders and non-responders based on recent KDIGO guidelines (proteinuria reduction after 6 months). There were 23 responders and 10 non-responders. The most important findings are the association of C5b-9 with proteinuria at time of diagnosis and during follow up. Further, uC5b-9 decreased during treatment only in those patients who responded to treatment, regardless of treatment applied. Further associations are shown by the authors.

Up to now studies analyzed the association of particularly urinary C5b-9 levels with crescents, histological scores, and kidney function, but data on longitudinal associations, especially under treatment is lacking. So, this aspect makes the study of Yu et al. definitely novel.

However, there are some points which should be addressed in more detail by the authors:

Results 3.5, page 6, line 184: It is not clear in this paragraph, which values did or did not correlate with TA proteinuria and or with eGFR during follow up. For example, the authors write: “Changes in corrected urinary C5b-9 levels were positively correlated with TA-proteinuria”, but further down the authors state: “corrected urinary C5b-9 levels did not correlate with either TA-proteinuria or the mean annual rate of eGFR decline”, this is confusing, please clarify.

Furthermore, the study lacks a comparison of the prognostic value of traditional factors, such as proteinuria or eGFR or hypertension with the novel biomarker candidate C5b-9. I would recommend performing a univariate and a multivariate analysis of factors at time of diagnosis, that predict response or non-response. The results should be compared with serum and urinary C5b-9 to show, if C5b-9 is only a bystander (to e.g. proteinuria) or an independent predictor (i.e. biomarker) which could be of value in future studies.

Author Response

Reviewer 2

However, there are some points which should be addressed in more detail by the authors:

Results 3.5, page 6, line 184: It is not clear in this paragraph, which values did or did not correlate with TA proteinuria and or with eGFR during follow up. For example, the authors write: “Changes in corrected urinary C5b-9 levels were positively correlated with TA-proteinuria”, but further down the authors state: “corrected urinary C5b-9 levels did not correlate with either TA-proteinuria or the mean annual rate of eGFR decline”, this is confusing, please clarify.

Thank you for your insightful comment. Per your suggestion, we have modified the sentence structure for clarity.

Results 3.5, page 7, line 225:

“Baseline serum and corrected urinary C5b-9 levels and changes in serum C5b-9 levels did not correlate with either TA-proteinuria or the mean annual rate of eGFR decline”

Furthermore, the study lacks a comparison of the prognostic value of traditional factors, such as proteinuria or eGFR or hypertension with the novel biomarker candidate C5b-9. I would recommend performing a univariate and a multivariate analysis of factors at time of diagnosis, that predict response or non-response. The results should be compared with serum and urinary C5b-9 to show, if C5b-9 is only a bystander (to e.g. proteinuria) or an independent predictor (i.e. biomarker) which could be of value in future studies.

Thank you for your constructive suggestion that we had not considered. We performed univariate and multivariate analyses of risk factors at the time of diagnosis, which predict response or non-response, in accordance with your suggestion. We have incorporated a description of this issue in the results and discussion sections.

Results, page 6, line 194:

“Based on logistic regression analyses between the prognostic value of baseline corrected C5b-9 levels with those of traditional prognostic markers, including eGFR, proteinuria, and hypertension (baseline corrected C5b-9, odds ratio 0.997; 95% confidence interval, 0.993–1.000; p = 0.078) (Table S4).”

Discussion, page 9, line 309:

“Although it showed higher significance than the other conventional risk factors, it was not a significant independent factor affecting treatment response. Considering that conventional risk factors were also not significant, it is thought that this was also because the sample size was too small for logistic regression analyses. If further research is conducted with larger cohorts and long-term follow-up durations, a significant relationship between urinary C5b-9 levels, eGFR, and treatment response may be obtained, and the current study might be used as a pilot study for such future studies.”

Supplementary Materials, page 10, line 375:

“Table S4. Univariable and multivariable logistic regression analyses of independent prognostic factors for treatment response.”

Round 2

Reviewer 1 Report

I appreciate the efforts by the authors to address my comments. I agree with the response of the authors

Author Response

Your suggestions have improved our manuscript. As you pointed out, this study is underpowered. The title and conclusion were changed to another term with weakened meaning. We thank you for the effort and time in evaluating our manuscript.

"Urinary C5b-9 as a Prognostic Marker in IgA Nephropathy"

"“Currently urinary C5b-9 is not a promising prognostic biomarker for IgAN, and further studies are needed.”

Reviewer 2 Report

The authors replied to all of my concerns in an adequate way. The revision resulted in new information, which is, that C5b-9 does not add any significant value compared to traditional risk factors for progression of IgAN, so – in a more stringent manner – it is not such a promising biomarker candidate, but rather an interesting finding. Furthermore, as discussed with Reviewer 1, this study is underpowered.

These new findings should result in some changes to the manuscript. Now, it is more an exploratory study and not so much late-breaking-news regarding a new biomarker. This should be mirrored by the wording.

Title: Please change to Urinary C5b-9 as a Prognostic Marker in IgA Nephropathy

Abstract: Include information that uC5b-9 was not better than hypertension and proteinuria and eGFR; as a conclusion: Further studies are needed, currently uC5b-9 is not a promising biomarker.

Table S4 should be moved into the manuscript.

Author Response

Reviewer 2

The authors replied to all of my concerns in an adequate way. The revision resulted in new information, which is, that C5b-9 does not add any significant value compared to traditional risk factors for progression of IgAN, so – in a more stringent manner – it is not such a promising biomarker candidate, but rather an interesting finding. Furthermore, as discussed with Reviewer 1, this study is underpowered.

These new findings should result in some changes to the manuscript. Now, it is more an exploratory study and not so much late-breaking-news regarding a new biomarker. This should be mirrored by the wording.

We thank you for the effort and time in evaluating our manuscript. We agree with your opinion.

Title: Please change to Urinary C5b-9 as a Prognostic Marker in IgA Nephropathy

Thank you for your insightful comment. As you suggested, we changed the title to Urinary C5b-9 as a Prognostic Marker in IgA Nephropathy.

Abstract: Include information that uC5b-9 was not better than hypertension and proteinuria and eGFR; as a conclusion: Further studies are needed, currently uC5b-9 is not a promising biomarker.

Thank you for your comment. We have incorporated a description of this issue in the abstract.

Abstract, page 1, line 24:

“Baseline urinary C5b-9 level was not significant independent factor that could predict the treatment response in logistic regression analyses (odds ratio 0.997; 95% confidence interval, 0.993 to 1.000; p = 0.078).”

Abstract, page 1, line 26:

“Currently urinary C5b-9 is not a promising prognostic biomarker for IgAN, and further studies are needed.”

Table S4 should be moved into the manuscript.

Thank you for your comment. In accordance with your suggestion, we changed Table S4 to Table 3 and showed it in the manuscript.

Results 3.4, page 6, line 214:

“Based on logistic regression analyses between the prognostic value of baseline corrected C5b-9 levels with those of traditional prognostic markers, including eGFR, proteinuria, and hypertension, no statistically significant independent factors could be identified that could predict the treatment response (Table 3).”
